# DEEP PROBABILISTIC TIME SERIES FORECASTING OVER LONG HORIZONS

## ABSTRACT

Recent advances in neural network architectures for time series have led to significant improvements on deterministic forecasting metrics like mean squared error. We show that for many common benchmark datasets with deterministic evaluation metrics, intrinsic stochasticity is so significant that simply predicting summary statistics of the inputs outperforms many state-of-the-art methods, despite these simple forecasters capturing essentially no information from the noisy signals in the dataset. We demonstrate that using a probabilistic framework and moving away from deterministic evaluation acts as a simple fix for this apparent misalignment between good performance and poor understanding. With simple and scalable approaches for uncertainty representation we can adapt state-of-the-art architectures for point prediction to be excellent probabilistic forecasters, outperforming complex probabilistic methods constructed from deep generative models (DGMs) on popular benchmarks. Finally, we demonstrate that our simple adaptations to point predictors yield reliable probabilistic forecasts on many additional problems of practical significance, namely large and highly stochastic datasets of climatological and economic data.

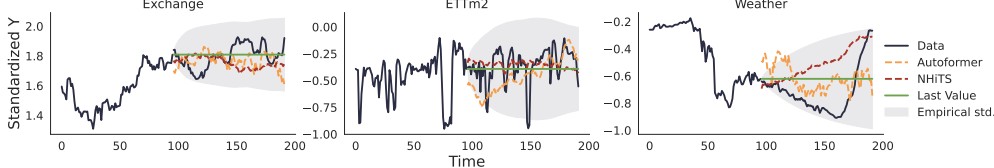

Figure 1: Example predictions on exchange rate (**left**), ETTm2 (a sequence of electricity transformer temperature readings, **center**), and weather (**right**) for NHiTS (Challu et al., 2022b), Autoformer (Wu et al., 2021), and last value predictions, as well as the historical standard deviation of the change from the last observed value. On the exchange (Lai et al., 2018) and ETTm2 (Zhou et al., 2021) datasets there is minimal structure to be exploited except on very short horizons, and forecasts tend to under-perform simple baselines. On semi-structured datasets like weather, models can capture some overall structure, such as NHiTS accurately predicting the final values in the forecasting window, but are still only on par with naive predictions. From these plots we see why probabilistic evaluation is necessary and point estimates are insufficient.

## 1 INTRODUCTION

Following deep learning's breakthroughs in sequence modeling for text and audio, significant research effort has sought to achieve comparable success in time series, where the unique challenges of data scarcity and long-range dependencies have created a niche for creative architecture design. In rapid succession, many new models and architectures have demonstrated improved point predictions on benchmarks adopted by the community (Challu et al., 2022b; Salinas et al., 2020; Wu et al., 2021). These methods hold incredible promise for real impact on time series based decision making, especially in economic domains that require highly accurate long-term predictions.

While demonstrating steady improvements, however, research on deep learning for point prediction frequently ignores a key but simple fact: the real world is complex and predicting the future accurately from past observations alone is often impossible. In highly structured time series, such as observed

Table 1: Multivariate results with varying prediction lengths. Bolded results indicate the best performing model, and italics the second best. In all cases simple statistics of the input data to the model are either the first or second best performing models in terms of both MSE and MAE accuracy. Historical Inertia (HI) (Cui et al., 2021) was also introduced as a trivial baseline but has worse performance than our constants.

| Models | Mean | | Last Value | | N-HiTS | | Autoformer | | Informer | | ARIMA | | HI | |
|---|---|---|---|---|---|---|---|---|---|---|---|---|---|---|
| Metric | MSE | MAE | MSE | MAE | MSE | MAE | MSE | MAE | MSE | MAE | MSE | MAE | MSE | MAE |
| Exchan. 96 | 0.139 | 0.269 | **0.081** | **0.196** | *0.092* | *0.211* | 0.197 | 0.323 | 0.847 | 0.752 | 0.296 | 0.214 | 0.112 | 0.251 |
| Exchan. 336 | 0.384 | 0.454 | **0.305** | **0.396** | *0.371* | *0.443* | 0.509 | 0.524 | 1.672 | 1.036 | 2.298 | 0.467 | 0.434 | 0.517 |
| Exchan. 720 | 0.938 | 0.736 | **0.823** | **0.681** | *0.888* | *0.723* | 1.447 | 0.941 | 2.478 | 1.310 | 20.666 | 0.864 | 0.955 | 0.772 |
| ETTm2 96 | **0.150** | *0.272* | 0.203 | 0.312 | 0.176 | **0.255** | 0.255 | 0.339 | 0.365 | 0.4536 | 0.225 | 0.301 | *0.158* | 0.261 |
| ETTm2 336 | **0.205** | **0.313** | *0.270* | *0.361* | 0.295 | 0.346 | 0.339 | 0.372 | 1.363 | 0.887 | 0.370 | 0.386 | 0.379 | 0.438 |
| ETTm2 720 | **0.261** | **0.350** | *0.335* | *0.401* | 0.401 | 0.426 | 0.422 | 0.419 | 3.379 | 1.388 | 0.478 | 0.445 | 0.446 | 0.477 |
| Weather 96 | *0.216* | 0.271 | 0.259 | *0.254* | **0.158** | **0.195** | 0.266 | 0.336 | 0.300 | 0.384 | 0.217 | 0.258 | 0.345 | 0.339 |
| Weather 336 | *0.313* | *0.336* | 0.377 | 0.338 | **0.274** | **0.300** | 0.359 | 0.395 | 0.578 | 0.523 | 0.330 | 0.347 | 0.529 | 0.440 |
| Weather 720 | *0.380* | *0.377* | 0.465 | 0.394 | **0.351** | **0.353** | 0.419 | 0.428 | 1.059 | 0.741 | 0.425 | 0.405 | 0.545 | 0.439 |

traffic, with strong periodicity from both daily and weekly cycles, we may be able to forecast point predictions with a high degree of accuracy. However, in *stochastic* time series which display only modest structure (e.g. periodicity or seasonality), such as precipitation or wind speed patterns, we cannot hope to produce accurate predictions of specific future outcomes using only historic observations.

Researchers working in architecture design for timeseries frequently overlook intrinsic stochasticity in benchmark datasets. In Figure 1 we show how even sophisticated methods struggle to forecast accurately on data with a low signal-to-noise ratio. In Table 1 we take this observation further and show that, shockingly, naive constant predictors outperform two state-of-the-art time series models (Lai et al., 2018; Zhou et al., 2021) on several widely reported MSE evaluations. The numbers shown are taken directly from Challu et al. (2022b), Wu et al. (2021), and Zhou et al. (2021), but include new columns showing the performance of simply predicting either the mean or the last value of the observations. Notably, these datasets span many domains of practical interest (finance, energy, and climatology), and contain varying levels of structure and periodicity.

*We present these surprising shortcomings of state-of-the-art models in order to encourage adoption of more meaningful evaluations. Although these deep learning methods are extremely good at extracting and extrapolating trends and periodic structure far into the future, aspiring to predict a constant value in the best case reflects a misalignment. Meaningful comparisons require a baseline that excels in both highly predictable and stochastic environments.*

Keeping in mind the need for probabilistic, rather than deterministic frameworks for forecasting, we instead ask if the strong trend extrapolation performance of point prediction models is being overlooked in the probabilistic time series literature. For datasets that have both noise and structure (e.g. wind), we find that high performing point predictors, such as NHiTS, typically outperform the mean forecasts provided by state of the art probabilistic methods. In Figure 2 we show the cumulative mean squared error for forecasts produced by NHiTS and the mean prediction taken from several popular probabilistic frameworks on common benchmark datasets as well as real world data.[1] Motivated by improving probabilistic forecasting while retaining the strength in trend extrapolation found in some deterministic models, we explore adapting point predictors to the probabilistic setting rather than building new models entirely from the ground up.

In particular, we examine two baselines for constructing high performing probabilistic forecasters, built on quantile regression and heteroscedastic variance models (Section 3). These methods leverage advances in architecture design for time series modeling without succumbing to the pitfalls of point prediction. In Section 4, we provide a detailed comparison of our models with state-of-the-art deep learning methods for probabilistic forecasting, demonstrating that recent advances in architecture design are directly relevant to uncertainty quantification. Finally, in Section 5, we demonstrate

---

[1] Models and datasets described in Sections 4 and 5 along with plots of cumulative CRPS scores.

Figure 2: Cumulative mean squared error with shade representing 2 standard deviations over 5 trials for both NHiTS and the mean prediction from several popular probabilistic forecasting models. NHiTS has better mean predictions than the probabilistic approaches in semi-structured settings like precipitation and wind speed modeling, where there are still underlying trends caused by seasonality and daily effects. To build better probabilistic models we aim to leverage the accurate trend capture abilities of models such as NHiTS.

our methods' exemplary performance on large and noisy datasets for climatology and financial forecasting.

We provide our code at https://anonymous.4open.science/r/timeformer-87D0.

## 2   RELATED WORK

**Insights from classical methods.**   Prior to the first applications of machine learning in forecasting (De Gooijer and Hyndman, 2006), time series methods grew out of signal processing. Exponential smoothing (Brown, 1959), ARIMA (Box and Jenkins, 1968), state space models (Kalman, 1960; Shumway and Stoffer, 1982), and the (G)ARCH families of models (Engle, 1982) use simple filters or autoregressive models to extract trend components from noisy data. Many modern forecasting methods move beyond fixed parametric representations in order to better extract complex structure from data, but signal processing fundamentals continue to be guiding principles behind approaches that use filtering to decompose inputs and construct predictions.

**Deep decomposition approaches.**   Many recent deep learning architectures provide generalizations of older seasonality accounting approaches using decompositions. N-BEATS (Oreshkin et al., 2019), for example, uses several layers removing trend and seasonality components from univariate times series, while N-HiTS (Challu et al., 2022b) combines this approach with a multi-scale forecasting setup that uses several different networks to predict at different scales. Similarly, Smyl (2020) uses RNNs with exponential smoothing for seasonality components, ultimately winning the M4 forecasting competition. Exponential smoothing has been a key factor in many recent architectures, such as Autoformer (Wu et al., 2021) and ETSFormer (Woo et al., 2022), which both use smoothing modules as part of transformer-inspired architectures. Wen et al. (2017) use sequence-to-sequence RNN and CNN models in conjunction with quantile regression to produce a multi-horizon quantile forecasting model.

As we demonstrated above, these methods have a critical but easily resolvable shortcoming: they are constructed and evaluated without uncertainty in mind. In order to investigate the true impact of developments and deep decomposition methods, we draw on recent work in probabilistic forecasting.

**Probabilistic forecasting.**   Salinas et al. (2020) proposed a deep autoregressive forecasting method with simple closed-form likelihood models. Other methods have combined deep autoregressive models with more complex conditional density estimators in an effort to better model the joint distribution over covariates. Salinas et al. (2019), for example, extend the popular non-parametric Gaussian copula process volatility (GCPV) model of Wilson and Ghahramani (2010) for high-dimensional forecasting with RNNs. More recently, authors have turned to using normalizing flows (Rasul et al., 2020; de Bézenac et al., 2020) and denoising diffusion models (Rasul et al., 2021; Tashiro et al., 2021) as highly flexible conditional density estimators. In particular, Rasul et al. (2020) and Rasul et al. (2021) condition normalizing flow and diffusion models on the outputs of an RNN to construct a joint model over all timesteps. In contrast, Tang and Matteson (2021) eschew conditioning on a deterministic backbone and uses a fully probabilistic transformer architecture.

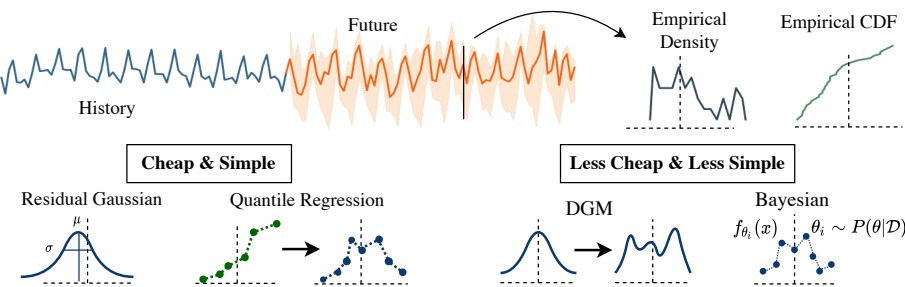

Figure 3: In probabilistic forecasting we form a distribution over possible future outcomes that most closely matches the empirical distribution at some future time point. Given a model architecture capable of mean trend extrapolation we can simply fit Gaussian distributions to the residuals, or simultaneously predict many quantiles of the distribution, as in quantile regression. These approaches are inexpensive relative to using a deep generative model (DGM) or the Bayesian posterior to quantifying uncertainty. We find, surprisingly, that residual Gaussians and quantile regression coupled with powerful point predictors often outperform more complex and expensive alternatives in the long horizon setting.

While prior work in probabilistic forecasting has focused on high-dimensional density estimation, we show that architecture design should be prioritized instead. Indeed, with extensive experiments, we demonstrate that when given a successful long-term predictor simple uncertainty representation is sufficient.

## 3 EXTENDING POINT FORECASTING METHODS

Prior work contains many methods for converting sequence models into probabilistic forecasters. We focus our attention on approaches that are *simple* and *computationally efficient* (Figure 4): quantile regression (Chernozhukov et al., 2010) and two-stage heteroscedastic Gaussian regression (Nix and Weigend, 1994). While quantile regression provides a more flexible forecast distribution that can asymmetric about the mean prediction, the Gaussian regression approach gives a closed form forecast which has practical use in applications like risk prediction.

**Quantile Regression** We first consider quantile regression (QR) for representing uncertainty because of its unique combination of simplicity and flexibility. Unlike simple parametric families, QR can learn arbitrary distributions while maintaining efficient sampling. QR learns a conditional order statistic, $f_\tau(x)$, given a quantile level $\tau \in (0, 1)$ using the pinball loss (Chernozhukov et al., 2010):

$$L_\tau(f_\tau, y) = (f_\tau - y)(\mathbb{I}\{y \le f_\tau(x)\} - \tau). \tag{1}$$

With $\tau = 0.5$, this loss reduces to mean absolute error, which yields an estimate of the conditional median. If we take $\tau = 0.01$ instead, the model yields an estimate of the smallest conditional percentile. To estimate a full predictive distribution, we train a series of conditional estimates for quantile levels $\{\tau_0, ..., \tau_k\}$ simultaneously (Chung et al., 2021; Rodrigues and Pereira, 2020; Tagasovska and Lopez-Paz, 2019). At test time, we use interpolation to ensure monotonic quantiles (Chernozhukov et al., 2010).

Quantile regression is popular instrument within the probabilistic forecasting community (Gouttes et al., 2021; Kan et al., 2022; Park et al., 2022; Romano et al., 2019), and we do not propose any novel extensions of it. Instead, we show that simple UQ methods can outperform more complex and computationally expensive techniques. When using quantile regression in conjunction with NHiTS we refer to the method as NHiTS-QR.

**Heteroscedastic Gaussian Regression** Distinct from NHiTS-QR, we propose to train a neural network to output the variance parameter in a Gaussian likelihood, analogous to earlier approaches (Bishop, 1995; Nix and Weigend, 1994; Rodrigues and Pereira, 2020). Here, we assume a het-

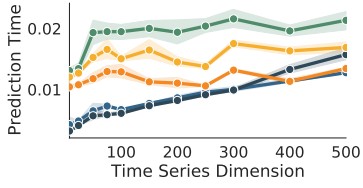 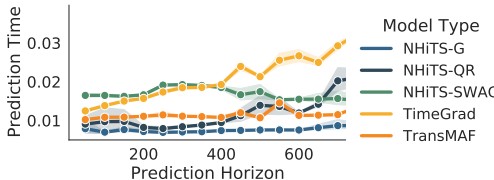

(a) Inference time vs Dimension.     (b) Inference Time vs Prediction Horizon.

Figure 4: **(a)** Time in seconds to generate a single forecast as a function of the output dimensionality of the time series. **(b)** Time in seconds to generate a single forecast as a function of prediction length. In both cases we see that the modified point predictor models (NHiTS-G and NHiTS-QR) run much faster than either diffusion-based models for all but the longest or highest dimensional forecasts.

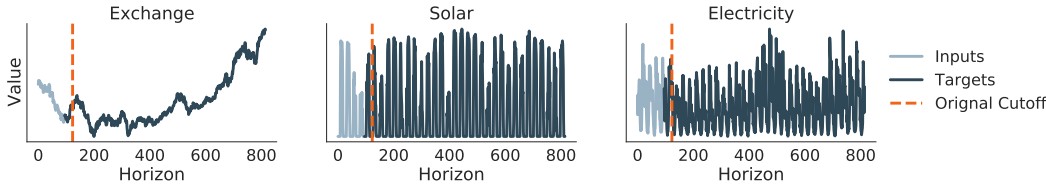

Figure 5: Many previous works on probabilistic time series forecasting have focused on short horizons, shown as the vertical dashed line. Here we extend the targets to more than 20 times the original length. By expanding both the forecast horizon and the test set size we able to validate methods on longer horizons as well as a broader distribution of test points.

eroscedastic Gaussian observation model, $\hat{y}_t \sim \mathcal{N}(f(x_{1:T})_t, g(f(x_{1:T})_t))$, where $f(\cdot)$ is a hierarchical time series model as described in Challu et al. (2022a) and $g(.)$ is a simple multi-layer perceptron. We term this model NHiTS-Gaussian (NHiTS-G). To train this model, we use maximum likelihood in a two-step manner, first training the mean model, $f(\cdot)$, using MSE loss, and then fixing that model and training the variance model, $g(\cdot)$, by maximizing the likelihood of the Gaussian observation model. Our approach parallels Model Agnostic Quantile Regression (MAQR) (Rodrigues and Pereira, 2020) and the heteroscedastic MLPs fit in Nix and Weigend (1994) and Bishop (1995), but with inductive biases particularly suited for probabilistic time series. While it is a strong assumption to assume the marginal distribution at each time point is Gaussian, this model serves as a simple baseline, and as we show in Section 4 is capable of outperforming more expensive methods.

## 4 BENCHMARK EXPERIMENTS

**Dataset Description** We consider multivariate time series datasets provided by the **GluonTS** time series library and used in several time series papers focused on uncertainty quantification (UQ) (Salinas et al., 2019; 2020; Tang and Matteson, 2021). There are notable shortcomings with this experimental setup, however, as the default formulations use limited test sets and short horizon predictions. For the exchange, solar, electricity, and traffic datasets the prediction horizon is only either 24 or 30 time steps, and the test sets are comprised of no more than *seven* testing forecast windows. These prediction horizons are less than 10% of the same evaluation sets as commonly used in the non-probabilistic time series literature, e.g. Challu et al. (2022a); Cui et al. (2021).

To test on these datasets more fully and compare the performance of probabilistic forecasters in more challenging and realistic settings we modify the testing and training splits from leaving approximately 3% of the data as the test set to be 80% train, 10% validation, 10% test in order to assess the performance across a set of long range horizons of 100's of steps into the future, with thousands of test points. We show examples of the expanded forecast windows in Figure 5, for further information on dataset setup, and additional experimental results see Appendix B.

**Metrics** For probabilistic forecasts, we compute CRPS (Gneiting and Raftery, 2007; Matheson and Winkler, 1976), summed over the dimensions of the time series, and averaged across the prediction

times. For a single prediction, the CRPS score is defined against the estimated cumulative distribution function (CDF), $\hat{F}$ as

$$\mathrm{CRPS}(\hat{F}, y) = \int_{\mathbb{R}} (\hat{F}(z) - \mathbb{I}_{(z-y)>0})^2 dz,$$

where $\hat{F}(z)$ is estimated as the empirical CDF produced by sampling from our forecasts. Given our forecast contains multivariate time series and is predicted out over multiple time steps, we compare using $\mathrm{CRPS}_{\mathrm{sum}}$, first by summing over the dimensions of the time series, then by averaging over time points. The summation gives $\hat{F}_{sum}(t)$ and $\mathbf{y}_t = \sum_i y_{t,i}$ where $y_{t,i}$ is the $d^{\mathrm{th}}$ dimension of time series $y$ at time $t$, and the average yields $\mathrm{CRPS}_{\mathrm{sum}} = \mathbb{E}_t \left[ \mathrm{CRPS}(\hat{F}_{sum}(t), \mathbf{y}_t) \right]$. $\mathrm{CRPS}_{\mathrm{sum}}$ is a proper scoring rule and takes into account both the sharpnes and accuracy of the prediction. CRPS can also be computed directly from the pinball loss (Eq. 1) as

$$\mathrm{CRPS}(F, y) = 2 \int_0^1 L_\tau(y, F^{-1}(\tau)) d\tau,$$

the integral of the pinball loss over all quantiles (Gneiting and Raftery, 2007; Laio and Tamea, 2007).

**Baselines** We include several probabilistic time series methods as baselines for comparison. These methods cover a breadth of approaches to time series forecasting including autoregressive models (Salinas et al., 2020), sequence to sequence based quantile regression (MQ-RNN and MQ-CNN) (Wen et al., 2017), and diffusion models TransMAF and TimeGrad (Rasul et al., 2021; 2020). These models represent state of the art and are highly competitive approaches on the widely used GluonTS datasets as seen in Rasul et al. (2021). Despite the strong performance shown by these methods for short horizon forecasting, at longer horizon predictions we see these established baselines suffer in performance, with the strong trend extrapolation provided by using an underlying deterministic model being critical to forecasting success.

### 4.1 PROBABILISTIC LONG TERM FORECASTING

While recent models such as TimeGrad (Rasul et al., 2021), TransMAF (Rasul et al., 2020), and DeepAR (Salinas et al., 2020) have achieved significant improvements in continuous ranked probability scores on benchmark datasets, these methods have primarily been explored in short forecasting settings with limited sized test sets. While the datasets typically contain thousands of observations in time, only the last few hundred are reserved for testing amounting in some cases to a mere $1\%$ of the data, and only in prediction intervals of 24 time steps, see Figure 13 for plots of the standard splits.

A primary strength of decomposition-based methods like Autoformer and NHiTS is their ability to produce long term forecasts, showing success with predictions out to 720 time steps in the future (Challu et al., 2022b; Wu et al., 2021). Our probabilistic forecasters build on these methods to be well suited for producing accurate long horizon mean forecasts, and ultimately strong probabilistic modeling due to the underlying strength of the point predictor.

**Long Range Evaluation** To more accurately gauge performance over long term forecasts with larger test sets, we change the GluonTS datasets to a $80\%/20\%$ train/test split, and predict on a horizons of $\{96, 192, 336\}$ time steps. We show the results in Figure 6.

While in highly stochastic settings with little trend to be extrapolated, such as the Exchange rate dataset, the advantages to using models like NHiTS-G and NHiTS-QR are muted. However as the datasets exhibit more structure, such as the daily effects in electricity consumption and traffic congestion, or the seasonal effects in solar radiation, the improved trend extrapolation given by the underlying point predictor leads to stronger performance. This trend is more significantly pronounced at longer forecast horizons, where we see NHiTS-G and NHiTS-QR achieving state of the art in longer term forecasting for the electricity, traffic, and solar datasets.

**Bayesian methods** In probabilistic time series forecasting we are primarily concerned with representing *epistemic uncertainty*, or uncertainty about the forecasts our model is producing (rather than aleatoric, or irreducible, uncertainty). If our aim is to accurately represent our model uncertainty without altering the base architecture then Bayesian methods are a natural choice (MacKay,

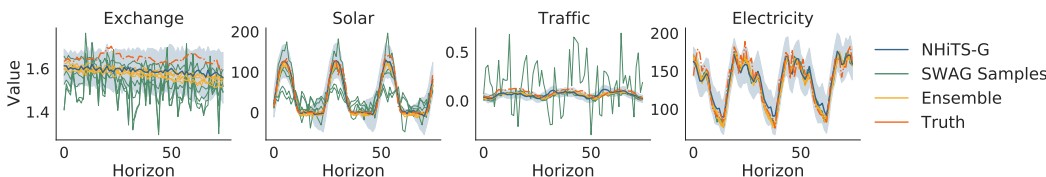

Figure 6: Barplot Of $\text{CRPS}_{\text{sum}}$ scores for varying prediction lengths. As the forecast horizon increases, NHiTS approaches consistently outperform current state of the art methods. Note, we used several hyper-parameters for TransMAF, including those given in publicly available implementation here, but were not able to find stable training procedures for all datasets. Error bars represent $\pm 2$ standard deviations over 5 random initializations, the $y$-axis is truncated for clarity.

Figure 7: Example forecasts produced by NHiTS-G in comparison to samples produced by SWAG and deep ensembles. In all cases the deep ensemble samples produce limited diversity and do not provide the uncertainty needed for probabilistic forecasting. SWAG samples tend to be noisy, and can fail to predict trends even when we are confident of their existence as on traffic.

1992), and recent advances in approximate Bayesian neural networks (BNNs) have given us simple approaches to build models with out of the box probabilistic forecasts (Maddox et al., 2019).

In Figure 7, we show uncertainties learned by NHiTS-G side by side with two different approximations of model uncertainty, Deep Ensembles (Lakshminarayanan et al., 2017) and SWAG (Maddox et al., 2019) for several datasets (Lai et al., 2018). On the traffic dataset, the base architecture is easily capable of producing accurate mean forecasts, leading to both ensembles and SWAG producing homogeneous forecasts that under-represent our true uncertainty. For the solar and traffic datasets we are able to forecast means that fit the cyclical nature of the data, but cannot predict peak solar radiation or traffic consistently leading to the need for uncertainty in our forecasts. In these cases SWAG overestimates our model uncertainty and fails to confidently predict the mean, even when it is appropriate to do so (such as predicting low solar radiation at night).

We show the cumulative CRPS of both the adapted point forecasting methods, as well as SWAG and deep ensembles in Figure 8. Despite promising performance on short horizons, as the length of the forecast increases the performance of SWAG and deep ensembles degrades, and both NHiTS based models tend to perform much better at long term predictions. While Bayesian methods might also provide a simple mechanism for wrapping uncertainty around an effective base model, they carry serious computational costs at test time, because we need to do as many forward passes through the model as there are samples, as illustrated in Figure 4.

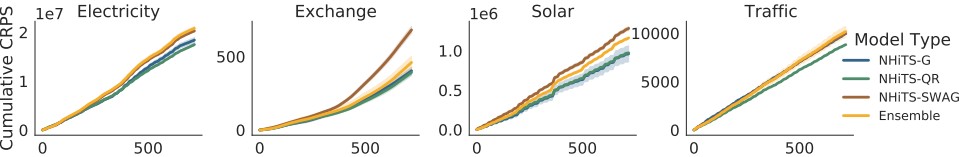

Figure 8: Cumulative CRPS of the NHiTS based methods with Bayesian baseline adaptations. While on short horizons SWAG and deep ensembles can provide reasonable forecast distributions, at longer horizons we find limited sample diversity or failure to predict trends leading to reduced performance.

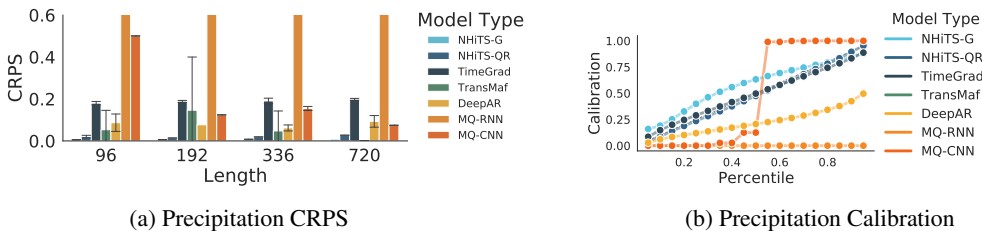

(a) Precipitation CRPS

(b) Precipitation Calibration

Figure 9: **Left:** CRPS scores for various prediction lengths in precipitation forecasting. **Right:** Calibration curves for forecasts of length 720. Both NHiTS-G and NHiTS-QR achieve low CRPS, outperforming all competing methods at every time horizon. While NHiTS-G attains lower CRPS than NHiTS-QR, NHiTS-QR produces highly calibrated forecasts. These metrics together suggest that methods like NHiTS-QR have practical relevance in climatological study, where accurate simulation is a key area of interest.

## 5   NOISY, LONG-HORIZON DATASETS WITH RICH STRUCTURE

Climatology and finance are key areas of interest for probabilistic forecasting, as they are stochastic domains where we cannot make confident predictions over long horizons, but retain exploitable structure like long term trends or cyclical climatological patterns. The ability to estimate distributions of climatological quantities of interest such as precipitation and wind speed has broad impact in ecological impact study, insurance pricing models, architectural and civil engineering, and beyond. Similarly, the probabilistic financial forecasting are central to the applications market pricing, automated trading, asset allocation. To demonstrate the efficacy of adapting deterministic models like NHiTS to probabilistic forecasting we examine predictive performance on two real world climatology datasets and historical prices from the component assets of the NASDAQ-100.

For these experiments we also consider calibration, as applications such as simulation and risk estimation demand that the probabilistic forecast is a faithful representation of the ground truth distribution at all quantiles. For a time series $y$ and forecast time $T$, to compute the calibration at $p$, $C_p$, we compute the empirical quantile of the forecast $q_T$ where $\hat{\mathbb{P}}(y_T < q_T) = p$. $C_p$ is then the observed frequency of $y_T < q_T$ over all forecasts and time horizons, yielding $C_p = \frac{1}{TK} \sum_k \sum_{t=1}^{T} \mathbb{1}_{y_{t,k} < q_{t,k}}$. If a predictor is calibrated, $C_p \approx p$ for all $p \in (0, 1)$, thus we can assess the overall calibration of a predictor by plotting calibration against percentiles, which should be close to the identity line.

**Precipitation Forecasting**   For precipitation forecasting we consider the United States Historical Climatology Network (USHCN) long-term daily climate records dataset (Menne et al., 2015). This dataset contains daily recorded precipitation values at approximately 1200 locations over the continental United States with observations ranging back over 100 years. In order to more accurately consider a scenario in which we are forecasting typical precipitation for a given day and location, rather than a single observed event, we preprocess the data by averaging over 5 day rolling windows.

In Figure 9b we show CRPS performance over varying horizons, as well as calibration for forecasts made over 720 time steps. Both NHiTS-G and NHiTS-QR attain low CRPS at all time horizons, and achieve high calibration relative to competing methods. While diffusion-based TimeGrad provides calibration on par with NHiTS-QR it yields much higher CRPS, particularly at the longer horizons that we are specifically interested in. MQ-CNN provides competitive performance in terms of CRPS for long forecasts, but gives very poor calibration, showing that the central quantiles of the forecast distribution are not well matched to the data.

**Wind Speed Forecasting**   For wind speed forecasting we use the United States Climate Reference Network (USCRN) dataset composed of wind speed observations taken at 5 minute intervals at 154 spatial locations in the United States for the full 2021 calendar year (Diamond et al., 2013). Wind speed forecasting provides a particularly challenging problem for forecasting models, as there are many cyclic trends in the data that relate to both seasonal and daily effects, but there are also gust-like effects that produce erratic localized behavior including random periods of either high or absent wind.

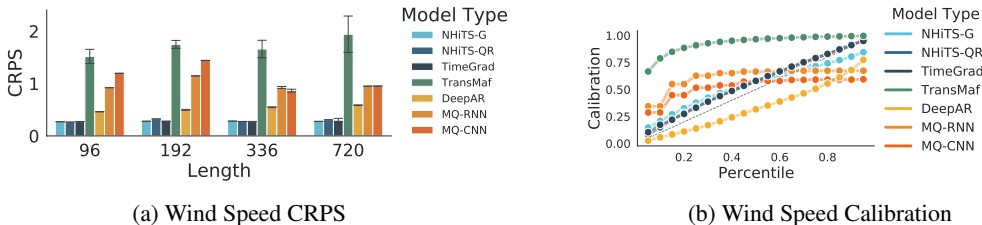

(a) Wind Speed CRPS

(b) Wind Speed Calibration

Figure 10: **Left:** CRPS scores for various prediction lengths in wind speed forecasting. **Right:** Calibration curves for forecasts of length 720. For all time horizons NHiTS-G, NHiTS-QR, and TimeGrad achieve low CRPS, with NHiTS-QR and TimeGrad both producing well calibrated forecasts. Wind forecasting presents unique challenges due to the stochastic nature of gust-like effects, and the ability of NHiTS-QR to produce calibrated forecasts indicates that we are accurately capturing both the frequency and magnitude of such effects.

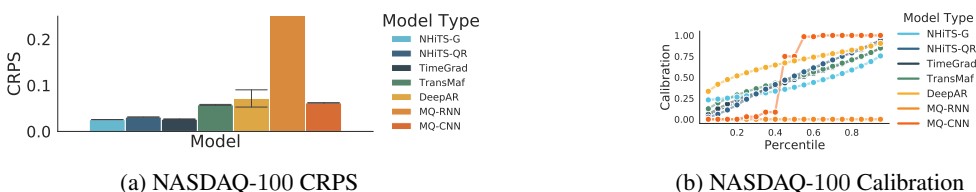

(a) NASDAQ-100 CRPS

(b) NASDAQ-100 Calibration

Figure 11: **Left:** CRPS scores for prediction lengths of 200. Error bars represent one standard deviation from the mean over 5 trials, and the plot is truncated on the $y$-axis for clarity. **Right:** Calibration curves. Both variants of the probabilistic adaptation of NHiTS achieve low CRPS values, however NHiTS-QR in particular achieves near perfect calibration. NHiTS-G underperforms in calibration as may be expected from the non-normality of securities prices.

Figure 10b includes both the CRPS and calibration comparison for the probabilistic NHiTS variant and competing methods. In wind speed forecasting TimeGrad performs approximately as well the adapted deterministic models in terms of both CRPS and calibration, but the remaining methods suffer both from higher CRPS scores and poor calibration.

**Financial Data**    To demonstrate the performance of the probabilistic forecasting for real world financial data we construct a dataset containing hourly data from the component assets of the NASDAQ-100 for the two years leading up to September 2022. In this setting we forecast 200 time steps into the future, which equates to approximately a month of hourly predictions when accounting for trading hours. We show in Figure 11b that again both NHiTS-G and NHiTS-QR are among the best performing models, with NHiTS-QR achieving nearly perfectly calibrated forecasts. Given the typically assumed log-normality of stock prices it is unsurprising that NHiTS-G is asymmetrically miscalibrated relative to its performance on the climatology datasets.

## 6    CONCLUSION

Time series has rapidly become a popular target of innovation in neural network architecture design, and we argue that this increased popularity must be accompanied by increased scrutiny of common baselines and evaluation methods. In particular, we demonstrate that probabilistic evaluation is critical for understanding the efficacy of new methods on highly stochastic data. Without probabilistic evaluation, many sophisticated methods can be matched or outperformed with naive predictors. In promoting probabilistic methods and evaluations, we also emphasize the importance of simple baselines for uncertainty quantification. From our experimental results, we conclude that architectural design can have a significant impact on the quality of learned forecasts. Contrary to a perceived consensus in recent work, we find that elaborate uncertainty quantification techniques are less important than architecture design. We hope that our work provides a foundation for future methods in probabilistic forecasting, especially those that target long horizons and noisy datasets.

## ETHICS STATEMENT

We do not anticipate that this work should have any negative societal implications. On the contrary, improvements in time series forecasting should have primarily beneficial impacts. For example, improvements in weather forecasting, especially at higher frequencies should provide longer windows for evacuation from severe weather such as hurricanes and tornadoes. Similarly, improvements in time series forecasting should be extremely helpful in disease progression forecasting, enabling doctors and other practitioners to have a better understanding of which patients need increased attention or different treatments (Jarrett et al., 2021). At the same time, over-reliance on machine learning models in quantitative finance settings (like in Section 5) could potentially lead to financial shocks and decreased robustness to risk in the stock market, like was the case for several of the first usages of statistical learning in these settings (Jorion, 2000; MacKenzie and Spears, 2014).

## REPRODUCIBILITY STATEMENT

Our experiments are easily reproducible with the scripts provided in our anonymized repo. The GluonTS and Autoformer benchmark datasets can be easily downloaded from each package's public code repo. The USHCN precipitation and wind speed datasets can be easily downloaded from public logs and are used without data cleaning. The hourly NASDAQ-100 dataset can also be easily reproduced using public logs and is used without data cleaning. The hyperparameters for our models are included in the appendix.

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

# Appendix

## Table of Contents

## A   LIMITATIONS

We see that our work, as a primarily empirical study, has several limitations:

- Our empirical work focuses primarily on well-used benchmark datasets that could be quite distinct from other time series forecasting problems used in the literature. Thus, approaches that do not perform well here might perform better on other tasks.

- Our specific architecture is limited to forecasting only up to a pre-specified number of steps into the future.

- Our training and evaluation procedure is quite distinct from earlier efforts for forecasting using online learning, preventing easy direct comparison with methods like ARIMA and Gaussian processes.

- If the underlying time series is extremely non-stationary and the input time series is quite different from any previously observed data, we may still be unable to extrapolate.

## B   EXPERIMENTAL DETAILS

### B.1   COMPUTE

Experiments were carried out across several computing platforms. All models were trained on a single GPU. Figures 1, 6, 7, 4 were generated using a single NVIDIA Titan RTX. The remaining figures were generating from experiments using a single GPU on a larger cluster with a mix of NVIDIA V100s, RTX 8000s, and A100s. Depending on hyperparameter setting and dataset, models take approximately 10-30 minutes to train.

### B.2   GLUONTS SPLITS

In Figure 13, we display the training and testing splits for the six GluonTS datasets, with the dividing line shown in orange. Only on Wikipedia and Taxi are the training and testing splits especially large given the quantities of the data. Furthermore, the responses on Wikipedia, solar, and electricity may not be well described as having Gaussian observation noise, as they roughly reflect counts or drop to zero at times (solar at night).

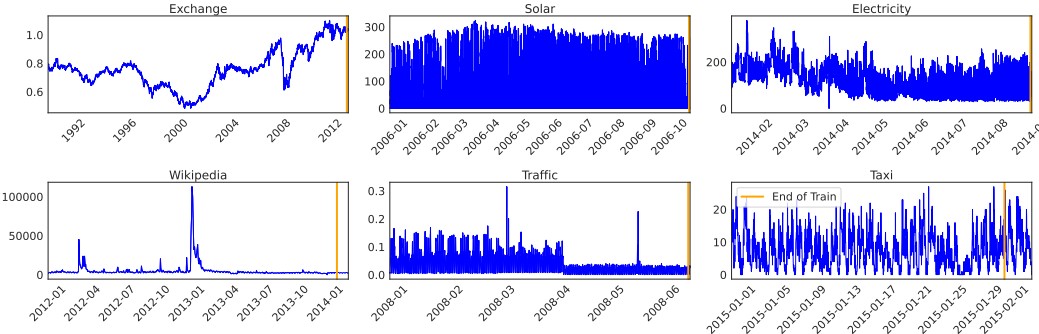

Figure 12: Training and test splits for GluonTS datasets. From top left down are exchange, solar, electricity, traffic, taxi, and wikipedia.

| Hyperparameter | Values |
|---|---|
| hidden dimension | $\{256, 512\}$ |
| # hidden layers | $\{2, 3\}$ |
| $\{\theta_1, \theta_2, \theta_3\}$ | $\{\{4, 2, 1\}, \{3, 2, 1\}, \{6, 2, 1\}, \{8, 2, 1\}, \{8, 4, 1\}\}$ |
| $\{\phi_1, \phi_2, \phi_3\}$ | $\{\{4, 2, 1\}, \{3, 2, 1\}, \{6, 2, 1\}, \{8, 2, 1\}, \{8, 4, 1\}\}$ |
| learning rate | $\{10^{-3}, 10^{-4}\}$ |
| weight decay | $2 \times 10^{-6}$ |
| quantiles (NHiTS-QR) | $\{0.01, 0.1, 0.2, 0.3, 0.4, 0.5, 0.6, 0.7, 0.8, 0.9, 0.99\}$ |

Table 2: Hyperparameters for NHiTS on short probabilistic evaluations

### B.3 DATA PRE-PROCESSING

Prior to training, all data were standardized using the `StandardScaler` function from Scikit Learn. Except where specified for the datasets taken from the GluonTS library from Alexandrov et al. (2020), we use the same data splits as Wu et al. (2021). For the long horizon experiments in Section 4.1, we use the first $10\%$ of the data as the validation set, the last $20\%$ as the test set, and the remaining data as the training set.

### B.4 NHiTS HYPERPARAMETERS

Table 2 shows the hyperparameter space we searched for the GluonTS results in Section 4.1.

### B.5 UNIVARIATE EXPERIMENT DETAILS

For the Autoformer and Transformer models, we used the public code Autoformer repo. We adapted the data loaders to work with GluonTS (Salinas et al., 2019) datasets (which have a history of 192 and prediction horizon of 24), and we searched over the common model hyperparameters from the repo, for both the Transformer and Autoformer architecture. For the univariate models, we reduced the models' capacity (through embedding dimension) from 512 to 256 to avoid overfitting. We converted multivariate tim eseries to univariate time series by sampling along the channels per batch and then flattening channels into the batch dimension. Models were trained for 20 epochs or until early stopping. We trained the NHiTS models with the hyperparameter space shown in Table 2.

## C FURTHER EXPERIMENTS

### C.1 COMPARISON WITH CONFORMAL QUANTILE REGRESSION

Conformalized quantile regression (CQR) (Romano et al., 2019) is a method for recalibrating a quantile regression forecaster using conformal sets constructed from a validation (calibration) dataset. In Figure 13, we show a comparison of our two probabilistic forecasting methods with CQR.

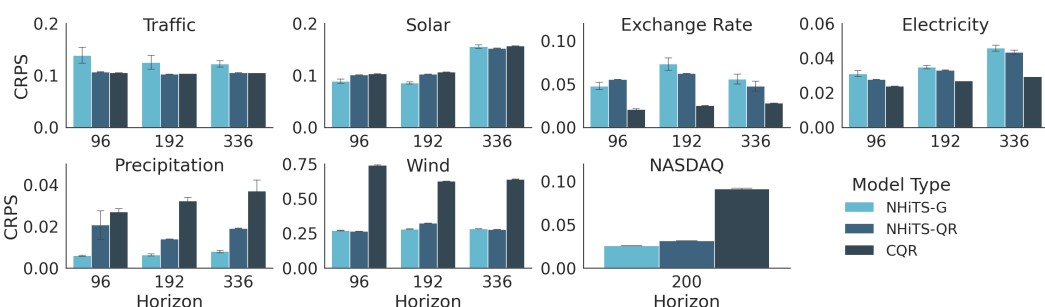

Figure 13: Comparing NHiTS-G and NHiTS-QR (ours) with conformal quantile regression (CQR), a conformal prediction method that adjusts the uncertainty estimates using a calibration/validation dataset. On the majority of the datasets conformal recalibration offers few benefits over our methods. In the datasets in the second row, CQR leads to significantly worse performance because constructing a compelling validation set is challenging.

