# OpenReview forum: "Deep Probabilistic Time Series Forecasting over Long Horizons"
_ICLR.cc/2023/Conference — Submitted to ICLR 2023_

### Official Review · Reviewer_RuZ4 · 2022-10-17

**Confidence:** 4
**Correctness:** 2
**Technical Novelty And Significance:** 2
**Empirical Novelty And Significance:** 2
**Recommendation:** 3

**Clarity, Quality, Novelty And Reproducibility:**

The paper is very clearly written. The novelty is poor given the similarity to quantile conformal regression which is not mentioned in this work or even just fitting simple likelihood on top of existing models. The reproducibility is good, the author provided code that should allow one to rerun the benchmarks (which I did not check nor try to run).


**Strength And Weaknesses:**

The main strength of the paper are that it is clearly written and has clear motivation since providing accurate probabilistic forecast on top of point forecast has many relevant applications.

The main weaknesses are as follow.

The approach is very related to conformal prediction, see [Romano 2019] for instance who proposes to fit a quantile regression model that can be guaranteed to be calibrated or [Hasson2021] who already showed the benefit of the approach by demonstrating how the top point forecast of M5 could be made a top competitor to the uncertainty track with a post-processing. Sadly, those approaches are not mentioned nor compared while being directly similar to the method proposed.

In addition, the method proposed does not distinguish itself clearly from just fitting a Gaussian/Quantile regression on top of an existing architecture. Indeed, the main improvement reported in the experiments seems to come from using an architecture better suited for long term predictions (NHiTS). However, one could also fit the same model with a Gaussian likelihood or a quantile regression as the loss function of NHiTS and likely obtain similar results in term of CRPS and calibration. The main difference of the approach proposed is the two step estimation but it is not compared to the simpler approach consisting on just fitting a distribution on top of an existing model and also have drawbacks that are not mentioned (two training must be done and latency will likely be twice higher).

To clearly justify their contribution, the authors should probably show that their methods outperforms both non conformal prediction approaches and fitting Gaussian/quantile regression approaches (the first one because it is state-of-the-art and the standard approach, the second one because it is arguably simpler and faster than the proposed method by the authors).

[Hasson 2021] Probabilistic Forecasting: A Level-Set Approach. Hasson et al. Neurips 2021.
[Romano 2019] Conformalized Quantile Regression. Romano et al. Neurips 2019.



**Summary Of The Paper:**

The paper proposes an approach to provide a probabilistic forecast on top of any point forecast. Two post-processing approaches in particular are investigated, one fits a Gaussian noise by learning a condition heteroskedastic variance on top of an existing point forecast, the other fits a quantile-regression model. Experiments are conducted by applying this post-processing to an existing method (NHits) on 7 datasets against several baselines with an emphasis on long-term forecasting performance.

**Summary Of The Review:**

Given the lack of novelty and lack of comparison with relevant approaches (conformal quantile prediction, likelihood directly fitted on top of previous models, [Hasson2021]), I recommend rejecting the paper in its current state.

Additional details:
* p3: Salinas et al. (2020) does not consider only Gaussian noise model but also other likelihood (neg-binomial for instance)
* p4: "To train this model, we use maximum likelihood in a two-step manner, first training the mean model ... using MSE loss, and then ... maximizing the likelihood of the Gaussian observation." This seems to be the claimed novelty but a clear baseline would to fit both mean/variance or have a quantile regression on top of an existing architecture. This would have twice lower training runtime and latency and should probably be added in the comparisons (in addition to conformal prediction).
* Figure 7: this figure is hard to read since you are plotting samples for some methods and confidence intervals for others. It would be much better to report calibration to allow to quantify the calibration of the probabilistic predictions as you are doing for wind/nasdaq
* why reporting calibration only on wind/nasdaq on not on other datasets (electricity/traffic/etc)? I would recommend reporting calibration plots on all datasets, calibration errors would also be important to provide quantitative metrics
* Figure 9: small comment calibration are typically plotted with equal axis as it is then easier to assess the quality of the fit

---

> ### Author Response · Authors · 2022-11-18
> **Response to Reviewer RuZ4**
>
> Thank you for the thoughtful review. Importantly, we do not claim to be focused on methodological novelty, rather we are emphasizing empirical findings that have been overlooked by the research community. By taking pre-existing solutions for both point predictions and adding probabilistic forecasts to deterministic models we are able to achieve state of the art on many practical problems.
>
> Thank you for the pointers to LSF and CQR, we will be sure to add appropriate citations in the camera ready. For our use cases LSF is unfortunately not practical, as it requires a distinct model to be trained for each prediction time step. As we are focused on long range predictions this becomes very costly and at test time is over 3000 times slower than any of the other methods tried, making it prohibitively slow to get comparisons at this time.
>
> You also suggest using conformalized quantile regression (CQR) as an alternative method. Conformal recalibration is in fact already an option that can be turned on in our codebase. We have included a new appendix section to show a comparison across the 7 datasets of Sections 4.1 and 5 to compare NHiTS-G, NHiTS-QR, and NHiTS-QR + CQR. In the Exchange Rate and Electricity datasets CQR provides some improvements, but in the Traffic and Solar datasets CQR has little impact. Importantly, in more nonstationary datasets where validation sets can be less reliable, like precipitation, wind speeds, and stock forecasting CQR leads to a degradation in performance over vanilla NHiTS-QR. We have included these results in the updated appendix.
>
> As you point out, our approach is no different from using a Gaussian likelihood or quantile regression loss in conjunction with NHiTS. In fact, that is precisely what our methods are, and therefore no comparison is necessary. While our primary model of interest is a combination of already discovered methods we do not claim novelty over this approach, rather the interest and novelty of our paper is in the empirical study. The empirical results of the paper show that despite not being explicitly developed for probabilistic forecasting (as opposed to models like TimeGrad and TransMAF), simple alterations to deterministic models achieve state of the art performance on long range forecasting problems.
>
> To your specific comments:
> - We use a two-stage approach to most directly leverage the underlying performance of NHiTS by starting with a pre-trained NHiTS model. Practically we find higher training stability by first fitting a vanilla NHiTS model then adding a simple model for uncertainty to that model rather than training against the joint objective of fitting a mean and a variance. In terms of training time and latency we notice very little impact. Indeed, the variance model is small and takes only a few epochs of training meaning the impact to training time is negligible, and as we show in Figure 4 NHiTS-G is one of the fastest models at test time despite being two-staged.
> - Figure 7: the choice of plotting samples vs confidence intervals is intentional, as models like SWAG and deep ensembles only provide samples and NHiTS-G only provides uncertainties. In the camera ready we will be sure to use the SWAG and deep ensemble samples to estimate confidence intervals and plot those as well for clarity.
> - Salinas et al.: Thank you for the pointer, we have corrected the sentence in the updated draft.
>
> Note we have also made a separate general post outlining our contributions.

---

> > ### Comment · Reviewer_RuZ4 · 2022-11-21
> > **answer to the authors**
> >
> > Thank you for your response and providing additional information regarding your paper. I acknowledge the points made in your rebuttal and I also checked the additional experiments on conformal predictions.
> >
> > I completely agree with you that novelty is not the only factor for including papers, however I believe the paper in its current form is still too short for ICLR.
> > While the points made in the paper are generally interesting (probabilistic forecasting methods struggles when horizon deteriorates compared to simple approach), this point is rather narrow as it only concerns (very) long horizon on top of probabilistic forecasting.
> >
> > That being said, I believe the paper could still be an interesting contribution but it would require a much better coverage of alternative approaches to provide a probabilistic extension on top of a point forecast. For instance,
> > * clearly spelling/measuring out the advantages of having a two phase training rather than fitting altogether the mean/variance of a likelihood (which I understand is the main contribution, but this comparison is not even provided)
> > * comparing with conformal predictions in a fair manner (I saw the results quoted, the improvements would probably not be statically significant given that 4 datasets are better/on par for CQR and 3 are worse), in addition no tuning of CQR is mentioned whereas I would assume that you tune your method
> > * LSF is discarded as being too slow for being included, while it could be the case you could still run it on the dataset that permits it. I am also not entirely convinced that the method would be unusable as you claim given that it consists in fitting tree base models which is probably decently fast and also given that the method was used on a set of 100K time-series (with shorten horizon but still).

---

### Official Review · Reviewer_5QvS · 2022-10-24

**Confidence:** 3
**Correctness:** 4
**Technical Novelty And Significance:** 4
**Empirical Novelty And Significance:** 4
**Recommendation:** 5

**Clarity, Quality, Novelty And Reproducibility:**

The reporting is clear and good quality, the approach is novel and innovative, and the relevant literature has been properly cited. Reproducibility has been described properly, and links to public repositories are provided. I did not manage to access these (due to anonymization of the submission?) and could not investigate the code itself. Some data sets may pose challenges for reproducibility: it is said that "the hourly NASDAQ-100 dataset can also be easily reproduced using public logs"; a code for doing this could improve the reproducibility further. It is not clear whether the code is provided with an open license.





**Strength And Weaknesses:**

Strength

+ intuitive representation of uncertainty through probabilistic treatment
+ scalability of the method
+ complements the current state of understanding in an innovative way
+ relevance to a broad variety real application cases

Weaknesses
- Limitations and weaknesses of the method could be addressed in the manuscript in more detail as these are relevant for applications

**Summary Of The Paper:**

This work considers long-term time series forecasting based on on deep probabilistic techniques. It shows that intrinsic stochasticity in data is often so remarkable that it poses notable challenges for the interpretation of many current deterministic methods. Importantly, the authors argue that uncertainty quantification is less important than neural network architecture design in many practical scenarios. The authors suggest a probabilistic evaluation framework that solves this challenge and improves prediction on common benchmarks and practical problems in climatology and economy.

**Summary Of The Review:**

Overall this is an interesting and timely contribution of good technical quality.

---

> ### Author Response · Authors · 2022-11-18
> **Response to Reviewer 5qvS**
>
> Thank you for the supportive review. We wish to reiterate that we are indeed focused on methods that are complementary with the current understanding of time series forecasting, and are broadly applicable to many domain areas. Importantly, these results and methods are compatible and complementary with a broad class of models, and should continue to be relevant even as research advances the state of both deterministic and probabilistic forecasting.
>
> We will be sure to include a more complete discussion of the weakness of our method in the camera ready. The main weakness of note is the reliance of the underlying point forecasting method (here NHiTS) on persistent patterns and trends in data. While this reliance can be a benefit, and is one of the reasons we see strong performance on structured and cyclic problems like the climate and traffic datasets, it may lead to underperformance in problems with very little exploitable structure.
>
> Finally, in the de-anonymized code release we will include an appropriate open source license. Regarding the financial data, we do include code for downloading the data from publicly available sources using the `yfinance` python package.

---

> ### Comment · Reviewer_5QvS · 2022-11-20
> **Updated assessment**
>
> After carefully revising the work and other review, I am lowering my score towards the other reviewers. In particular, I agree about the limitations in methodological novelty that have been brought up.

---

> > ### Author Response · Authors · 2022-11-20
> > **Please clarify?**
> >
> > Would you please kindly clarify? In our general post we have emphasized that our contributions are primarily not about methodological novelty. Do you believe that a radically new method should be required for ICLR acceptance?

---

### Official Review · Reviewer_Cp5J · 2022-10-24

**Confidence:** 4
**Correctness:** 2
**Technical Novelty And Significance:** 1
**Empirical Novelty And Significance:** 2
**Recommendation:** 3

**Clarity, Quality, Novelty And Reproducibility:**

This paper is ambiguous on how its conclusion is reached. There is also not too much technical novelty. The authors have provided code to reproduce their experiments.

**Strength And Weaknesses:**

Strength:
The authors have conducted extensive empirical evaluations and conducted a comprehensive background review. The most inspiring part is the discussion on the impact of intrinsic stochasticity and the structure of data on forecasting models. Based on this property, the authors have divided the analysis into time series from different domains. This can serve as an interesting reference for the characteristics of time series datasets and provide insights on which model to choose.

Weakness:
This paper is driven by pure empirical studies without theoretical insights or methodological innovation. The employed forecasting models and loss functions are all well-studied. The major work done by the authors is to combine them together and make a comparison on different datasets. Moreover, the authors don't provide a clear explanation of why the NHiTS is chosen as the base model and what is the intuition of its better performance than other complex methods. Lastly, I am also confused about how the authors reached the conclusion that architecture design should be prioritized over uncertainty quantification since the whole paper is comparing CRPS results across different methods. The authors apparently need a transition on this.

**Summary Of The Paper:**

This paper provides a new view of the probabilistic forecasting problem. The authors have shown some drawbacks of some current state-of-the-art probabilistic forecasting methods and propose that a simple model coupled with quantile loss or heteroscedastic Gaussian regression can often outperform more complex and expensive alternatives, particularly in the long horizon setting. The authors support their claim on multiple time series datasets from different domains and with distinct properties. They have also studied how the prediction horizon and time series dimension affect the inference time. Finally, they reach the conclusion that the architecture design of forecasting models should be prioritized over uncertainty quantification techniques.

**Summary Of The Review:**

This paper is an empirical study of probabilistic forecasting models, while it lacks solid technical novelty and theoretical insights. There is not much innovation in experiments either. I think this paper will benefit from another round of revision.

**Update after rebuttal**

The authors' responses have addressed some of my concerns, but I believe they are still not sufficient to significantly change my view of this paper. I would encourage authors to incorporate answers to the abovementioned questions in the next revision and discuss the underlying theoretical insights on why the methodology is performed.

---

> ### Author Response · Authors · 2022-11-18
> **Response to Reviewer Cp5j**
>
> Thank you for the thoughtful review. Indeed, we are focused on the practical impacts of stochasticity in time series modeling, and believe that the contributions of the paper are significant to the broader community. Namely, demonstrating through clear empirical study, that simple but overlooked methods in time series modeling are more than capable of state of the art performance in domains where we cannot make confident point predictions.
>
> To that end, we base our approach on NHiTS due to its success in deterministic forecasting, with leading performance across many benchmarks. In principle however, our findings are model agnostic, and will hopefully be adapted to future methods developed for point predictions as research advances. We will be sure to clarify both the motivation for using NHiTS and our perception of the future of our work in the camera ready.
>
> Lastly, we do not emphasize architecture design over uncertainty quantification. As you note we are focused on calibration- and uncertainty-based comparisons such as CRPS. We instead propose prioritizing architecture design over sophisticated techniques for density estimation, as we repeatedly find that the proper deployment of good architectures with simple uncertainty forecasting methods is capable of outperforming more sophisticated probabilistic models, such as TimeGrad or TempFlow.
>
> Note we have also made a general post outlining our contributions.

---

### Author Response · Authors · 2022-11-18
**General Remark to Reviewers and AC**

We would like to thank all reviewers for their comments. We emphasize that the paper makes several fundamental and timely contributions:
- We show that in widely used deterministic evaluations of time series forecasting models, including major AutoGluon benchmarks, highly sophisticated methods can be outperformed by naive predictions.
- We refocus evaluation on long term probabilistic extrapolation, which provides a more meaningful measure of what regularities and global structure can be extracted from the data.
- We also demonstrate the importance of modeling stochasticity in these settings.
- By adapting high performing deterministic models to the probabilistic setting we provide a simple and scalable approach to uncertainty modeling for time series forecasting, achieving state of the art performance on long range evaluation tasks.

While we really appreciate the feedback and thoughtful comments, we believe the significance of these contributions, their potential for impact, and especially the bigger picture context, have been overlooked.

There is a pressure in conference review cycles to propose sophisticated “highly novel” methods, which are often never adopted in practice, because similar results can be achieved by much simpler harder-to-publish alternatives. It is our contention that the observations in this paper should have more value than yet another new method with good performance on standard benchmarks, and that novelty should not be narrowly construed. Indeed, our findings suggest that there are real issues with the way time series forecasting methods are being evaluated.

---

### Comment · Area_Chair_vZUg · 2022-11-19
**Please respond to author feedback**

Dear reviewers,

The authors have provided their feedback. Please respond, and at least acknowledge you've read them.

Best, AC

---

### Decision · Program_Chairs · 2023-01-20

**Decision:**

Reject

**Justification For Why Not Higher Score:**

While I think large-scale empirical studies should be welcomed at ICLR, these type of papers need to provide rock-solid results and good insights to be considered for acceptance. From reviews, unfortunately this paper, while promising, is still not quite there yet in its current form.

**Justification For Why Not Lower Score:**

N/A

**Metareview: Summary, Strengths And Weaknesses:**

This paper presented a simple approach and multiple experiments in order to support their main claim in the paper: with simple adaptations high performing deterministic models can be made into state of the art probabilistic forecasters.

The main issue pointed out by the reviewers is the technological novelty of the proposed simple adaptation approach. In reply, the authors did not object to this criticism, but instead they emphasised on the need of careful evaluation of the methodologies.

To be clear, most of the reviewers welcomed this type of study of testing whether we really need complex methods to make long-term time series forecasting work, for which I completely agree with them. Rock-solid works on empirical evaluations with good insights are welcomed for presentations at ICLR.

However, even by considering the contribution on large-scale empirical evaluation side, reviewers found the results to be promising but not complete. Feedbacks for improvements include: adding suitable baselines and making the comparison fairer, having a better narrative to connect the many different experiments, and show a coherent insight about why the simpler method can perform better.

I would encourage the authors to improve their presentation and organisation of the many experiments, and make suitable amendments regarding experimental comparisons following reviewers' recommendations.

**Summary Of Ac-Reviewer Meeting:**

N/A